# ^18^F-Facbc in Prostate Cancer: A Systematic Review and Meta-Analysis

**DOI:** 10.3390/cancers11091348

**Published:** 2019-09-11

**Authors:** Riccardo Laudicella, Domenico Albano, Pierpaolo Alongi, Giovanni Argiroffi, Matteo Bauckneht, Sergio Baldari, Francesco Bertagna, Michele Boero, Giuseppe De Vincentis, Angelo Del Sole, Giuseppe Rubini, Lorenzo Fantechi, Viviana Frantellizzi, Gloria Ganduscio, Priscilla Guglielmo, Anna Giulia Nappi, Laura Evangelista

**Affiliations:** 1Department of Biomedical and Dental Sciences and of Morpho-functional Imaging, Nuclear Medicine Unit, University of Messina, 98125 Messina, Italy; riclaudi@hotmail.it (R.L.); sergio.baldari@unime.it (S.B.); 2Department of Nuclear Medicine, University of Brescia and Spedali Civili Brescia, 25123 Brescia, Italy; Doalba87@libero.it (D.A.); francesco.bertagna@unibs.it (F.B.); 3Unit of Nuclear Medicine, Fondazione Istituto G.Giglio, 90015 Cefalù, Italy; pierpaolo.alongi@hsrgiglio.it (P.A.); gloria.ganduscio@gmail.com (G.G.); 4Department of Health Sciences, University of Milan, 20142 Milan, Italy; giovanni.argiroffi@gmail.com (G.A.); angelo.delsole@unimi.it (A.D.S.); 5Nuclear Medicine Unit, IRCCS Policlinico San Martino, 16132 Genoa, Italy; matteo.bauckneht@gmail.com; 6Nuclear Medicine Unit, AO Brotzu, 09134 Cagliari, Italy; micboero@gmail.com (M.B.); priscilla.guglielmo@yahoo.it (P.G.); 7Department of Radiological Sciences, Oncology and Anatomical Pathology, Sapienza University of Rome, 00161 Rome, Italy; giuseppe.devincentis@uniroma1.it; 8Nuclear Medicine Unit, Department of Interdisciplinary Medicine, University of Bari Aldo Moro, 70124 Bari, Italy; giuseppe.rubini@uniba.it (G.R.); anna.giulia.nappi@gmail.com (A.G.N.); 9Department of New Technologies and Translational Research in Medicine and Surgery, Nuclear Medicine Unit, University of Pisa, 56126 Pisa, Italy; fantechi.lorenzo@gmail.com; 10Department of Molecular Medicine, Sapienza University of Rome, 00185 Rome, Italy; viviana.frantellizzi@uniroma1.it; 11Nuclear Medicine Unit, Veneto Institute of Oncology IOV - IRCCS, 35128 Padua, Italy

**Keywords:** prostate cancer, ^18^F-FACBC, PET/CT, recurrence, meta-analysis, review

## Abstract

Trans-1-amino-3-^18^F-fluorocyclobutanecarboxylic-acid (anti-[^18^F]-FACBC) has been approved for the detection of prostate cancer (PCa) in patients with elevated prostate-specific-antigen following prior treatment. This review and meta-analysis aimed to investigate the diagnostic performance of ^18^F-FACBC positron emission tomography/computed-tomography (PET/CT) in the detection of primary/recurrent PCa. A bibliographic search was performed including several databases, using the following terms: “FACBC”/“fluciclovine” AND “prostate cancer”/“prostate” AND “PET”/“Positron Emission Tomography”. Fifteen and 9 studies were included in the systematic reviews and meta-analysis, respectively. At patient-based analysis, the pooled sensitivity and specificity of ^18^F-FACBC-PET/CT for the assessment of PCa were 86.3% and 75.9%, respectively. The pooled diagnostic odds-ratio value was 16.453, with heterogeneity of 30%. At the regional-based-analysis, the pooled sensitivity of ^18^F-FACBC-PET/CT for the evaluation of primary/recurrent disease in the prostatic bed was higher than in the extra-prostatic regions (90.4% vs. 76.5%, respectively); conversely, the pooled specificity was higher for the evaluation of extra-prostatic region than the prostatic bed (89% vs. 45%, respectively). ^18^F-FACBC-PET/CT seems to be promising in recurrent PCa, particularly for the evaluation of the prostatic bed. Additional studies to evaluate its utility in clinical routine are mandatory.

## 1. Introduction

Prostate cancer (PCa) is the most frequently detected type of cancer in men and constitutes a major healthcare problem in developed countries [1], remaining the second most common cause of cancer-related death in the Western world [2].

Following initial diagnosis, the majority of men receive several treatments, such as usually a radical prostatectomy ± lymphadenectomy or radiation/brachytherapy in case of localized disease, and systemic therapy in case of widespread disease. Relapse remains common despite advances in primary treatment and improved overall survival (OS) with a biochemical recurrence developing in 20% to 40% of patients [3,4,5,6].

The management of primary and recurrent PCa patients has been completely changed after the inclusion of new imaging modalities, such as magnetic resonance imaging (MRI) and positron emission tomography (PET). MRI is a well-documented method to evaluate the extension of the primary tumor and to detect and localize recurrent cancer within the prostate [7,8,9]. However, routine multiparametric (mp) MRI is still limited by its poor specificity to differentiate significantly from indolent PCa [10].

In the last 10 years, PET/computed tomography (PET/CT) has gained an important role in the evaluation of patients with PCa. Radiolabeled choline PET/CT has demonstrated the ability to detect the presence of early recurrence of disease when conventional imaging resulted negative [11]. Furthermore, the recent introduction of radiolabeled prostate specific membrane antigen (PSMA), like ^68^Ga-PSMA and ^18^F-PSMA, has significantly improved the detection rate, also in case of early recurrence of disease (such as a prostate-specific antigen (PSA) <0.5 ng/mL) [12].

Trans-1-amino-3-^18^F-fluorocyclobutanecarboxylic acid (anti-[^18^F]-FACBC) is an amino acid PET tracer that has shown to be promising for visualizing PCa. This tracer was developed for L-amino acid transport evaluation; it demonstrated favorable dosimetry with the liver being the critical organ [13]. Its safety, tracer stability, and uptake kinetics in patients have been reported in a phase I trial [14]. Nowadays, ^18^F-FACBC is approved by the Food and Drug Administration (FDA) and the European Commission (EC) to detect PCa in patients with elevated PSA following prior treatment. Approval was based on encouraging diagnostic performance and histologically confirmed data on patients with biochemical recurrence [15]. Recently it was included in the National Comprehensive Cancer National (NCCN) guidelines for the management of recurrent PCa patients.

Until now, few pooled data have been published about the role of ^18^F -FACBC PET/CT in patients with PCa. Ren et al. [16] collected data from six studies, published between 2011 and 2014 and including 251 patients that concluded for a good sensitivity of ^18^F -FACBC PET/CT for the detection of PCa recurrence. In 2015, Yu et al. [17] published a critical analysis of the available tracers for PET/CT in PCa, collecting data for ^18^F -FACBC from five studies (n = 84 subjects), showing a limited detection rate of this imaging technique for the recurrence of post-prostatectomy PCa (detection rate = 40%). However, in May 2016, ^18^F -FACBC PET/CT received the approval by the Food and Drug administration for use in patients with suspected recurrent PCa [18]. In the last years, many prospective and retrospective experiences have been performed, and therefore, a new update of the recent findings seems necessary, not only in the restaging but also in the initial staging of disease.

Therefore, the present review and meta-analysis aimed to investigate the diagnostic performance of ^18^F -FACBC in the detection of primary and recurrent PCa patients.

## 2. Materials and Methods

### 2.1. Search Strategy and Study Selection

A bibliographic search until 30 April 2019 was performed by including the following databases: Pubmed, Scopus, Embase, Web of Science, Cochrane library, and Google Scholar. The terms used were “FACBC” or “fluciclovine” AND “prostate cancer” or “prostate” AND “PET” or “Positron Emission Tomography”. The search was carried out with and without the addition of filters (such as English language only; type of article: original article, research article; subjects: humans only). Three reviewers (Domenico Albano, Viviana Frantelizzi and Matteo Baucknhet) performed the literature search, and two independent reviewers (Priscilla Guglielmo and Lorenzo Fantechi) selected the study inclusion and data extraction in duplicate. Any discrepancies were resolved by a consensus, when necessary. All recognized records were combined, and the full texts were retrieved. Full texts were further evaluated by four reviewers (Giovanni Argiroffi, Riccardo Laudicella, Pierpaolo Alongi and Laura Evangelista). Moreover, a search across the databases was completed by another reviewer (Anna Giulia Nappi) checking the references of the studies included to further improve the eligibility. 

This systematic review was carried out using established methods [19], and the presentation of results was made according to the PRISMA guidelines [20]. All studies that fulfilled the inclusion criteria were considered eligible for the systematic review and meta-analysis: (a) a sample size more than 10 patients; (b) the index test: ^18^F-FACBC PET/CT; (d) the outcomes, such as detection rate (DR), true positive (TP), true negative (TN), false positive (FP), and false negative (FN), which allowed us to construct 2 × 2 contingency tables. Moreover, in the case of studies that included the same population, the report with the highest number of enrolled patients was considered for the meta-analysis. Conversely, reviews, clinical reports, meeting abstracts, and editor comments were excluded. The quality assessment included both the risk of bias assessment and applicability concerns by using QUADAS-2 evaluation [21].

### 2.2. Data Extraction

For each included study, general information was retrieved, such as basic data (authors, journal, year of publication, country and study design), patient characteristics (number of patients, mean or median age, Gleason score), type of treatment, mean or median PSA value at PET time, and PSA kinetic values.

### 2.3. Statistical Method

StatsDirect and Meta-Analyst (version Beta 3.13; [22]) were used to carry out the analysis. Heterogeneity was tested using the χ^2^ and the I2 tests. The χ^2^ -test provided an estimate of the between-study variance and the I2 test measured the proportion of inconsistency in individual studies that cannot be explained by chance. According to Higgins et al. [19], the values of 25%, 50%, and 75% for heterogeneity (I2) were considered low, moderate, and high, respectively. In accordance with the recommendation of the Cochrane Oral Health Group, the meta-analysis was carried out with the random-effect model as the number of studies was equal or superior to 4. 

Data on diagnostic performance such as pooled sensitivity, pooled specificity, positive and negative likelihood ratio (LR+ and LR−), diagnostic odds ratio (DOR) with 95% confidence intervals (CIs) for the evaluation of primary and recurrent PCa, were assessed. A patient-based and a region-based meta-analyses were carried out in accordance with available data. Publication bias was assessed using a funnel plot. A symmetrical plot was indicative of the absence of publication bias.

## 3. Results

### 3.1. Search Results

The literature search revealed 40 articles published from 1 January 2007 to 30 April 2019. Reviewing titles and abstracts, we excluded 24 articles because these did not fit with the field of interest or because these papers were letters, editorials, reviews or due to the patient data overlap. Therefore, 15 studies were selected and included in the systematic reviews and 9 articles were considered for the meta-analysis (Figure 1). Also the papers by the developers of ^18^F-FACBC were considered [23,24].

### 3.2. Study Characteristics

The basics characteristics of the included studies are reported in Table 1 [15,23,24,25,26,27,28,29,30,31,32,33,34,35,36]. The number of enrolled patients ranged from 15 to 596, and a total of 1226 PCa patients were included. The selected articles were published by researchers from Europe, USA, and Japan. Four studies were retrospective whereas 11 studies were prospective. ^18^F-FACBC PET/CT was performed in the preoperative setting in 6 studies (*n* = 178 patients), for the detection of recurrence in patients with biochemical relapse after primary treatments in 8 studies (n = 1033 patients) and in both settings in 1 study (*n* = 15 patients). In the restaging, the mean value of PSA ranged between 0.44 and 17.94 ng/mL.

The mean and median age of the patients ranged from 42 to 90 years. The Gleason score (GS) was ≤6 in 49 (4%) patients, 7 in 376 (30.6%) patients, ≥8 in 142 (11.6%) patients, not available in the remaining 659 (53.8%). No significant adverse effects after the administration of ^18^F-FACBC were reported.

### 3.3. Methodological Quality

All 15 studies were evaluated qualitatively using the QUADAS-2 tool (Appendix A; Figure 2). The risk of bias was unclear for patient selection in 1 study, which did not provide information regarding consecutive enrollment [15]. For the index test and reference standard, the risk of bias was low in 6 studies [24,29,31,32,33,36]. For flow and timing, many studies reported time intervals between PET/CT examinations and pathological or other imaging confirmations. The applicability of the included studies was adequate in the majority of reports, being unclear only in 1 study for the reference standard [30].

### 3.4. Qualitative Results

PET/CT was employed in 14/15 studies, without CT contrast media injection, whereas PET/MRI was used in 2 studies [31,32]. The injected radiopharmaceutical activity and the time between radiotracer injection and image acquisition were similar across all studies.

Analysis of PET images was mostly performed using visual analysis; however, additional semi-quantitative criteria, i.e., maximal standardized uptake values (SUVmax), was performed in some reports [23,26,27,28]. ^18^F-FACBC PET/CT or PET/MRI identified the presence of PCa in prostatic and extra-prostatic bed, such as in the regional, distant lymph nodes and bone. The DR was available in 9/15 studies. It ranged between 36% and 90%, being different in accordance with PSA serum levels (Table 2). Andriole et al. [34] demonstrated that DR was broadly proportional to pre-scan PSA: lesions were detected in 79% patients with PSA ≥ 1.0 ng/mL and in 84% with PSA ≥ 2.0 ng/mL. On the other side, some authors found that there was no statistically significant difference in the PSA values and PSA doubling-time (PSAdt) between patients with positive and negative findings [26,35]. England et al. [35] reported that the DR was significantly higher for patients with GS > 7 than those with a score equal to 7.

The performance of ^18^F-FACBC PET/CT was different based on the phase and the site of PCa (Table 3). In particular, in the initial staging, the sensitivity for the primary and lymph nodes metastasis was 71% [32] and 67% [36], respectively. In the restaging setting, the sensitivity for the prostatic bed and extra-prostatic bed recurrence was 89% [24] and 90% [15], respectively. Interestingly, in the study by Turkbey et al. [25], ^18^F-FACBC uptake in tumors was similar to that in benign prostatic hyperplasia (BPH). However, Jambor et al. [32] reported that SUVmax in the primary tumor was statistically significantly higher for patients with GS > 7 than GS = 6 or BPH, thus underlying the importance of the patient selection.

Akin-Akintayo et al. [33] compared ^18^F-FACBC PET/CT with mpMRI in patients with recurrent PCa showing a higher detection for the first modality (overall 94.7% vs. 36.8%); Turkbey et al. [25], instead, performed a sector-based comparison with histopathologic analysis in patients with a recent diagnosis of PCa, revealing lower sensitivity and specificity for ^18^F-FACBC PET/CT than for T2-weighted imaging (67% and 66% vs. 73% and 79%, respectively), but combined modalities achieved a positive predictive value of 82% for tumor localization, which was higher than that with either modality alone. Another study proved higher positivity rates with ^18^F -FACBC PET/CT than enhanced CT at all PSA levels, PSAdt and GS in patients with suspected recurrent PCa [29]. Furthermore, the performance of ^18^F-FACBC PET/CT was superior to those of ^111^In-capromab SPECT/CT regarding sensitivity for prostatic and extra-prostatic bed (89% vs. 69% and 100% vs. 10%, respectively) [24]. Finally, two studies directly compared ^18^F-FACBC with ^11^C-Choline PET/CT, demonstrating a greater detection rate for ^18^F-FACBC than 11C-Choline, either on a patient- and a lesion-based analysis and despite the PSA serum levels [27,28]. 

The change of management with ^18^F-FACBC PET/CT was reported by Andriole et al. [34], in 122 out of 213 patients (56%); the most frequent change was to withhold planned salvage or non-curative systemic therapy in favor of watchful waiting. Moreover, Akin-Akintayo et al. [30] demonstrated that ^18^F-FACBC PET/CT was able to modify the radiotherapy field and overall radiotherapy decision in 40.5% of patients with post-prostatectomy recurrent PCa.

### 3.5. Quantitative Results

In accordance with the inclusion criteria, the quantitative assessment was available in 9 studies [15,23,24,26,29,31,32,33,36] (Table 4). At patient-based analysis (n = 6 studies), the pooled sensitivity and specificity of ^18^F-FACBC PET/CT scan for the assessment of primary and recurrent PCa were 86.3% (95% CIs: 79.6–91.4%) and 75.9% (66.9–83.5%) with an heterogeneity of 78.6% and 88.7% (both *p* <0.0001), respectively. Moreover, the pooled DOR value was 16.453 (95% CI: 5.241–51.646), with heterogeneity of 30%. At the regional based-analysis (n = 4 studies), the pooled sensitivity of ^18^F-FACBC PET/CT for the evaluation of primary and recurrent disease in the prostatic bed was higher than that in the extra-prostatic regions (90.4% vs. 76.5%, respectively); conversely, the pooled specificity was higher for the evaluation of extra-prostatic region than the prostatic bed (89% vs. 45%, respectively). Furthermore, LR+ was high in the extra-prostatic region, while LR- was low in prostatic bed, with heterogeneity of 0%. No asymmetry in the forest plot was found; therefore, no publication bias was present across the studies.

## 4. Discussion

As previously mentioned, the meta-analysis from Ren et al. [16] reported that 18F-FACBC PET/CT had a high sensitivity (pooled sensitivity = 87%) and a moderate specificity (pooled specificity = 66%), therefore it can be considered an useful non-invasive, metabolic imaging technique for the diagnostic workup of PCa relapse. In the present meta-analysis, performed in 1226 PCa, the pooled sensitivity and specificity were 86% and 76% respectively, thus showing a slight increase for the specificity.

Furthermore, in the analysis by Yu et al. [17], FACBC showed a detection rate ranged between 22% and 61% for prostatic disease and between 19% and 33% for extra-prostatic disease, in accordance with the primary treatments (radical prostatectomy or radiotherapy). In our meta-analysis, we did not evaluate the pooled detection rate, but we calculated the pooled sensitivity and specificity. As illustrated in Table 4, the sensitivity of 18F-FACBC was equal to 90% for the identification of disease in the prostatic bed and 77% for extra-prostatic organs.

However, in the last years, PSMA-PET has rapidly been introduced in clinical practice for the management of patients with recurrent PCa, particularly in case of low PSA levels [37]. Already, the study by Yu C-Y et al. [17] reported that 18F-FACBC, Choline and Acetate-PET have similar detection rate for overall site of disease after radical prostatectomy or radiotherapy (ranged between 40% and 81%), but PSMA was able to reach a detection rate ranged between 82% and 96% in the same setting.

Two recent papers about a head-to-head comparison between 18F-FACBC and 68Ga-PSMA PET/CT have been published. The data are controversial. In the study by Pernthaler et al. [38] involving 58 patients with recurrent PCa with a PSA level ranged between 0.2 and 230 ng/mL, 18F-FACBC detected more accurately the presence of a local recurrence than 68Ga-PSMA, due to its favorable biodistribution. Furthermore, the authors found that 18F-FACBC is almost equivalent to 68Ga-PSMA-11 in detecting distant metastases of PCa recurrence. Conversely, in the study by Calais et al. [39] enrolling 50 patients with recurrent PCa, the detection rate of PSMA-PET was significantly higher than 18F-FACBC (56% vs. 26%, respectively) in case of a PSA level <1 ng/mL. However, the authors found that the detection rate for the local recurrence was higher for 18F-FACBC than 68Ga-PSMA PET/CT (38% vs. 14%, respectively). The missing data about the diagnostic performance, in terms of sensitivity and specificity in both the above-mentioned papers, represent a great limitation for the final conclusion on “the best radiopharmaceutical agent”. A recent paper by Lawhn-Heath et al. [40] reported that the sensitivity and specificity of 68Ga-PSMA-11 for recurrent PCa are equal to 89.1% and 31.2%, thus registering a high rate of false positivity.

From the present systematic review and meta-analysis arise some considerations:
^18^F-FACBC is more performant than ^111^In-capromab SPECT/CT and ^11^C-Choline for the detection of PCa recurrence. Therefore, if available it should be preferred in patients with a PSA increase, after primary treatments. However, data about the comparison with ^18^F-Choline PET/CT are missing and should be explored, also considering the radioisotope properties. The combination of ^18^F-FACBC PET/CT with mpMRI (or with a PET/MRI) seems useful for the detection of primary PCa, and therefore, it would be suggested in case of undetectable tumors in patients with a negative biopsy but a persistent PSA level increase. However, the interpretation of this sophisticated imaging required a great experience and a significant learning curve.The sensitivity for the evaluation of lymph node metastasis in the initial staging of disease is moderate (45%–66%; [31,36]), like for the other radiopharmaceuticals (radiolabeled PSMA and Choline; [41,42]). Probably the recent introduction of new imaging modalities, such as digital PET/CT or PET/MRI that has a higher spatial resolution, would improve the pathological lymph node detection.The pooled sensitivity for the identification of recurrence in prostate bed is high, being >90% with a limited pooled specificity (about 45%), probably due to the FP findings in case of inflamed cells, as reported by Oka et al. [43]. However, the absent uptake of radiopharmaceutical in the bladder represents a great advantage for the identification of peri-anastomotic PCa recurrence. Further data about the complementary role of ^18^F-FACBC and MRI are required for the assessment of prostatic bed recurrence, at different PSA levels.The recurrence in the extra-prostatic site may be assessed by ^18^F-FACBC PET/CT with a moderate sensitivity and specificity, independently from the PSA levels. However, the correlation with PSA kinetics is warranted in a selected large cohort of patients, thus testing the final impact on the patient management.Despite some articles have defined a potential impact of ^18^F-FACBC PET/CT on therapeutic management, there is still a lack information with regard to its role in radiotherapy planning and other adapted therapy.

## 5. Future Researches

More data about the correlation between the detection rate of ^18^F-FACBC PET/CT or PET/MRI and the PSA kinetics are warranted, particularly by a site and lesion-based analysis. The complementary role of ^18^F-FACBC PET/CT and mpMRI for the evaluation of the prostatic bed should be largely explored. A head-to-head comparison with ^18^F-Choline would be used in order to definitely assess its advantages in clinical routine. Data about the utility of ^18^F-FACBC PET/CT in patients undergoing or not hormonal therapy are required. The evaluation of response to therapy (chemotherapy or new hormonal agents) by ^18^F-FACBC PET/CT should be assessed. Finally, additional data about the effect of ^18^F-FACBC PET/CT on patient management is required, by considering both PSA levels and histopathological PCa characteristics.

## 6. Conclusions

^18^F-FACBC PET/CT seems to be promising in recurrent PCa, particularly for the evaluation of the prostatic bed. However, additional studies are mandatory in order to evaluate its utility in clinical routine.

## Figures and Tables

**Figure 1 cancers-11-01348-f001:**
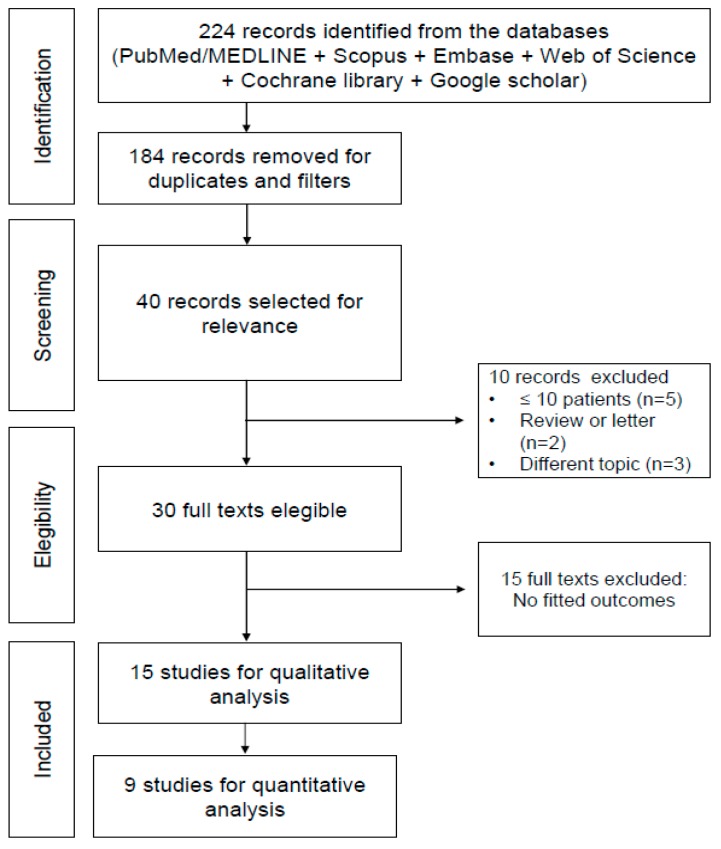
PRISMA flow-chart.

**Figure 2 cancers-11-01348-f002:**
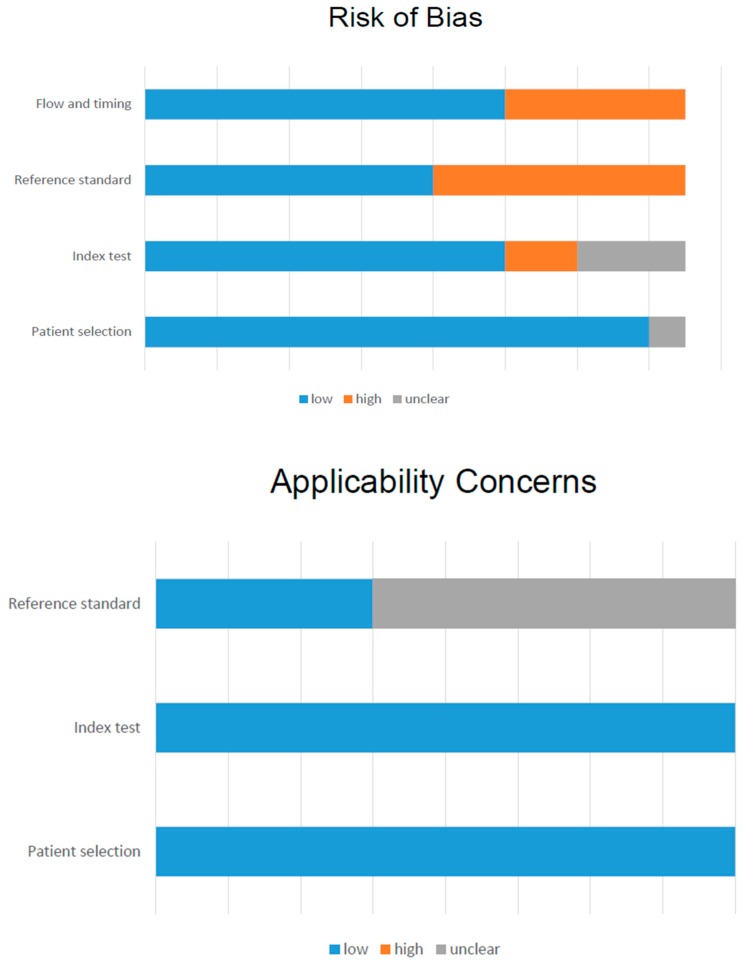
QUADAS 2 score of all included studies.

**Table 1 cancers-11-01348-t001:** Characteristics of selected studies.

Study Characteristics						Patient Characteristics					
Authors	Year	Journal	Country	Study Design	Setting	N. pts	Mean Age (Range)	Gleason Score (n)	Type of Treatment (*n*)	Mean PSA (Range)	Mean PSA Doubling Time (Range)
Schuster et al. [23]	2007	JNM	USA	Prospective	Staging (*n* = 9) Restaging (*n* = 6)	15	62y (45–76)	6 (2) 7 (2) 8 (2) 9 (2) 10 (1) NA (6)	SP (1), BCT + RT + CTR (1), BCT (2), BCT + RT (1), RP + RT (1), naive (9)	15 ng/mL (1.9–71)	NA
Schuster et al. [24]	2011	Radiology	USA	Prospective	Restaging	50	68.3y (50–90)	NA	RP (13), CTR, HFUS, EBRT, and/or BCT (37)	6.62 ng/mL (0.11–44.74)	NA
Turkbey et al. [25]	2014	Radiology	USA	Prospective	Staging	21	62y (44–73)	6 (3) 7 (12) 8 (5) 9 (1)	RARP + LND (21)	13.5 ng/mL (3.55–37.3)	NA
Kairemo et al. [26]	2014	BioMed Research Intern	Finland	Retrospective	Restaging	26 *	68.1y (56–77)	5 (3) 6 (7) 7 (7) 8 (3) 9 (5)	RP + RT (12), RT (13), ADT (20), BT (11), CHT(5), ^153^Sm-EDTMP (7), Denosumab (1)	7.9 ng/mL (0.11–69)	positive FACBC 3.2mo (0.3–6) negative FACBC 31.2mo (8–84)
Nanni et al. [27]	2014	ClinGenitourin Cancer	Italy	Prospective	Restaging	28	67y (55–78)	6 (1) 7 (16) 8 (6) 9 (4) 10 (1)	RP (28), RT (11), ADT (14)	2.9 ng/mL (0.2–14.6)	NA
Nanni et al. [28]	2015	ClinNucl Med	Italy	Prospective	Restaging	50	67y (55–78)	≤6 (4) 7 (31) 8–10 (15)	RP (50), RT (23), ADT (21)	3.2 ng/mL (0.24–15.6)	NA
Odewole et al. [29]	2016	EJNMMI	USA	Retrospective	Staging	53	67.57y (49–90)	7 (49) NA (4)	RP (7), EBRT (5), BCT (6), CTR (4), HT (1), 2 or more treatment (30)	7.2 ng/mL (0.11–44.8)	18.6mo ^##^ (−31.6–357.8)
Bach-Gansmo et al. [15]	2017	J Urol	Norway Italy USA	Retrospective	Restaging	596	67y (42–90)	6.7 (110) ^§^ 7.4 (355) ^§§^	RP (130), RP + other but no RT (62), RT (76), RT + other (266), other but no RT/RP (41)	5.43 ng/mL (0.05–82.0)	NA
Akin-Akintayo et al. [30]	2017	ClinNucl Med	USA	Prospective	Restaging	42	62y (42–75)	7 (42) ^#^	RP (42)	2.1 ng/mL (0.07–11.15)	NA
Selnaes et al. [31]	2018	EurRadiol	Norway	Prospective	Staging	26	66.2y (55–71.9)	7 (11) 8 (8) 9 (7)	RARP + LND (26)	14.6 ng/mL (3.7–56.9)	NA
Jambor et al. [32]	2018	EJNMMI	Finland	Prospective	Staging	26	65y ** (49–76)	6 (1) 7 (17) 8 (2) 9 (6)	RARP + LND (26)	12 ng/mL (4.1–35)	NA
Akin-Akintayo et al. [33]	2018	Eur J Radiol	USA	Prospective	Staging	24	70.8y (60–83)	7 (24) ^#^	BCT (3), RT (3), PT (1), CTR (1), CTR + HT (1), BCT + other treatment but no RP (13), other treatment but no BCT (2)	8.5 ng/mL (2.2–29.3)	NA
Andriole et al. [34]	2019	J Urol	USA	Prospective	Restaging	213	66.4y (46–90)	≤6 (27) 7 (134) ≥8 (50) NA (2)	RP (121), RP + RT (43), EBRT (21), BCT (1), EBRT + BCT (2), EBRT + ADT (17), EBRT + CTR (2), CTR (1), BCT + ADT (1), EBRT + BCT + ADT (2), HIFU (1), High-dose BCT (1)	4.24 ng/mL (0.2–93.5)	NA
England et al. [35]	2019	Clin Nucl Med	USA	Retrospective	Restaging	28	67.1y (53–77)	7 (19) 8 (3) 9 (6)	Primary treatment RP (22), RP+ EBRT (3), RP + EBRT + ADT (1), EBRT + ADT (2) Salvage therapy RT (6), ADT (1), RT + ADT (1), LND (1)	0.44 ng/mL (0.1–1.0)	6.38mo (1.6–16.8)
Suzuki et al. [36]	2019	Japanese J Clin Oncol	Japan	Prospective	Staging	28	67.9 (57–77)	<6 (1) 7 (12) 8 (8) 9 (8)	RARP + LND (28)	17.94 ng/mL (1.20–82.38)	NA

RP = radical prostatectomy; RS = radical surgery; EBRT = external beam radiotherapy; RT = radiotherapy; ADT = androgen deprivation therapy; LND = lymph nodal dissection; HT = hormone therapy; RARP = robot assisted radical prostatectomy; BT = bisphosphonate therapy; CHT = chemotherapy; BCT = brachitherapy; CTR = criotherapy; HFUS = high-frequency ultrasound; SP = subtotal prostatectomy; PT = proton therapy; NA = not-available. * 1/26 patient was affected by meningioma, considered as negative; ** Median value of the initial 32 patients; ^§^ Median Gleason-score value in Recurrent Prostate Cancer; ^§§^ Median Gleason-score value in Primary Standard of Truth; ^#^ Median Gleason-score value; ^##^ Only for 49/53 patients.

**Table 2 cancers-11-01348-t002:** The selection of the studies.

Author, (Ref)	Year	Journal	Country	N pts	Outcome	DR	TP	TN	FP	FN
Schuster et al. [23]	2007	JNM	USA	9	Accuracy LN (patient-based)	NA	2	5	0	2
Schuster et al. [24]	2011	Radiology	USA	50	Accuracy (PB) FACBC (region-based)	NA	32	8	4	4
					Acc (extra-*p*) FACBC (region-based)		10	7	0	0
					Acc (PB) Capromab (region-based)		25	7	5	11
					Ac (extra-*p*) Capromab (region-based)		1	7	0	9
Turkbey et al. [25]	2014	Radiology	USA	21	DR for primary	19/21 (90%)				
					Lesion-based		33	0	38	15
					Accuracy MRI (les-based)		34	0	21	14
Kairemo et al. [26]	2014	BioMed Research Intern	Finland	26 **	DR	17/26 (65%)				
					Patient-based		11	12	3	0
Nanni et al. [27]	2014	ClinGenitourin Cancer	Italy	28	DR (comparison with Choline)	10/28 (36%)	NA	NA	NA	NA
Nanni et al. [28]	2015	ClinNucl Med	Italy	50	DR (comparison with Choline)	17/50 (34%)	NA	NA	NA	NA
Odewole et al. [29]	2016	EJNMMI	USA	53	DR (all PSA levels and clinical data)	41/53 (77.4%)				
					Accuracy (PB) FACBC		31	9	7	4
					Accuracy (PB) CT		4	14	2	31
					Accuracy (extra-pr) FACBC		12	15	0	15
					Accuracy (extra-pr) CT		3	15	0	23
Bach-Gasmo et al. [15]	2017	J Urol	Norway Italy USA	596	DR	403/595 (67.7%)				
					Lesion-based		153	216	93	91
					Region-based (PB)		74	14	20	10
					Region-based (Extra-prost)		36	1	3	4
					Patient-based		98	14	21	10
Akin-Akintayo et al. [30]	2017	ClinNucl Med	USA	42	DR (change in radiotherapy strategy)	34/42 (81%)	NA	NA	NA	NA
Selnaes et al. [31]	2018	EurRadiol	Norway	26	Accuracy for LN	NA				
					Patient-based	NA	4	16	0	6
					Region-based	NA	6	185	0	14
Jambor et al. [32]	2018	EJNMMI	Finland	26	Accuracy LN	NA				
					Patient-based		7	19	0	0
					Region-based		NA	NA	NA	NA
Akin-Akintayo et al. [33]	2018	Eur J Radiol	USA	24	Accuracy (PB) FACBC *	NA	13	1	8	0
					Accuracy (PB) MRI *		5	5	4	8
					Accuracy (extra-*p*) FACBC		7	9	1	1
					Accuracy (extra-*p*) MRI *		4	7	3	4
Andriole et al. [34]	2019	J Urol	USA	213	DR (also for PSA level)	122/213 (57%)	NA	NA	NA	NA
England et al. [35]	2019	ClinNucl Med	USA	28	DR (for site and clinical data)	13/28 (46%)	NA	NA	NA	NA
Suzuki et al. [36]	2019	Japanese J ClinOncol	Japan	28	Accuracy LN	NA				
					Patient-based		4	19	3	2
					Lesion-based		4	28	5	3

DR = detection rate; NA = not available; LN = lymph node; PB = prostatic bed; * M1 reader; ** 1/26 patient affected by meningioma was considered as negative.

**Table 3 cancers-11-01348-t003:** Accuracies based on the study setting and the type of analysis.

Type of Analysis	Study Name (Year), Ref	Setting (Site)	TP	FN	TN	FP	Sensitivity	Specificity
Patient-based analysis	Suzuki et al. (2019), [36]	Staging (LN)	4	2	19	3	66.6%	86.3%
	Selnaes et al. (2018), [31]	Staging (LN)	4	6	16	0	45%	80.8%
	Jambor et al. (2018), [32]	Staging (primary)	7	0	19	0	70.6%	82.8%
	Bach-Gasmo et al. (2017), [15]	Restaging (all)	98	10	14	21	90.7%	40%
	Kairemo et al. (2014), [26]	Restaging (all)	11	0	12	3	76.2%	68%
	Schuster et al. (2007), [23]	Staging/restaging (all)	2	2	5	0	50%	66.7%
Region-based analysis (PB)	Schuster et al. (2011), [24]	Restaging	32	4	8	4	88.9%	66.7%
	Bach-Gasmo et al. (2017), [15]	Restaging	74	10	14	20	88.1%	41.2%
	Akin-Akintayo et al. (2018), [33]	Staging	13	0	1	8	78.3%	31.6%
	Odewole et al. (2016), [29]	Staging	31	4	9	7	88.6%	56.3%
Region-based analysis (extra-PB)	Schuster et al. (2011), [24]	Restaging	10	0	7	0	75%	70.6%
	Bach-Gasmo et al. (2017), [15]	Restaging	36	4	1	3	90%	25%
	Akin-Akintayo et al. (2018), [33]	Staging	7	1	9	1	87.5%	90%
	Odewole et al. (2016), [29]	Staging	12	15	15	0	45.9%	80%

LN = lymph node; TP = true positive; FN = false negative; TN = true negative; FP = false positive.

**Table 4 cancers-11-01348-t004:** The pooled diagnostic performance for ^18^F-FACBC (independently from the clinical setting and site).

Meta-Analysis Results	Patient-Based Analysis (95% CI)		Region-Based Analysis (PB) (95% CI)		Region-Based Analysis (ex-PB) (95% CI)	
	Value	I2	Value	I2	Value	I2
**Pooled sensitivity, %**	86.3% (79.6–91.4%)	78.6%	90.4% (84.8–94.4%)	22.1%	76.5% (66–85%)	87.3%
**Pooled specificity, %**	75.9% (66.9–83.5%)	88.7%	45.1% (33.2–57.3%)	63.3%	88.9% (73.9–96.9%)	78.7%
**DOR**	16.453 (5.241–51.646)	29.9%	8.026 (3.841–16.769)	3.5%	24.820 (3.777–163.12)	36%
**LR+**	4.557 (1.685–12.324)	72.9%	1.598 (1.088–2.349)	70%	6.024 (0.568–63.943)	85.6%
**LR−**	0.337 (0.166–0.681)	63.6%	0.221 (0.130–0.375)	0%	0.251 (0.058–1.090)	71.6%

PB = prostatic bed; PPV = positive predictive value; NPV = negative predictive value; DOR = diagnostic odds ratio; LR = likelihood ratio; IC = interval of confidence; I2 = inconsistency.

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
