# Peer review of "^18^F-Facbc in Prostate Cancer: A Systematic Review and Meta-Analysis"

_cancers, 2019, doi:10.3390/cancers11091348_

Round 1
Reviewer 1 Report
This review article is aimed to describe current status and clinical potential of anti-18F-FACBC for detection of prostate cancer. The paper is well organized and contains scientific information.
However, now we are going to PSMA-PET era. The authors provided only limited information about the difference between FACBC-PET and PSMA-PET. Could you add more information about this point?
Author Response
CANCERS-559403
Tile manuscript: 18F-FACBC IN PROSTATE CANCER: A SYSTEMATIC REVIEW AND META-ANALYSIS
The authors are thankful to the Reviewers for their comments. As requires, some updated information about the comparison of 68Ga-PSMA and 18F-FACBC has been added. Moreover, the English has been revised and other improvements have been given.
Please below you can find the point-by-point response (in red) to the queries.
REVIEWER 1
This review article is aimed to describe current status and clinical potential of anti-18F-FACBC for detection of prostate cancer. The paper is well organized and contains scientific information.
Q1. However, now we are going to PSMA-PET era. The authors provided only limited information about the difference between FACBC-PET and PSMA-PET. Could you add more information about this point?
R1. A new paragraph about the comparison between FACBC-PET and PSMA-PET has been added as a sub-part of the discussion paragraph.
Reviewer 2 Report
18F-FACBC is an FDA approved radiopharmaceutical for prostate cancer imaging. It is anticipated to be more prevalent in our clinical practice due to its potential to patients with biochemical recurrence following treatment.
One of the concerns about the application of 18F-FACBC is its moderate specificity/performance at low PSA cases. Extensive studies in association with other feasible options are needed to validate this point.
This manuscript is a systematic review and a meta-analysis to investigate the sensitivity and specificity of 18F-FACBC at the patient-based and regional-based levels.
In my review, the provided list of publications on this is informative while the main questions about applications of 18F-FACBC cannot be answered due to lack of enough data to be analyzed. My major concern about this paper is similarly of this study and its outcomes with the following papers:
1- Ren, Jingyun, et al. "The value of anti-1-amino-3-18F-fluorocyclobutane-1-carboxylic acid PET/CT in the diagnosis of recurrent prostate carcinoma: a meta-analysis." Acta radiologica 57.4 (2016): 487-493.
2- Yu, Chung Yao, et al. "Comparative performance of PET tracers in biochemical recurrence of prostate cancer: a critical analysis of literature." American journal of nuclear medicine and molecular imaging 4.6 (2014): 580.
None of the above paper has been cited in this manuscript and the “Discussion” section is only a list of highlighted information.
Unfortunately, I can not recommend the current version of this manuscript to be considered for publication.
Author Response
CANCERS-559403
Tile manuscript: 18F-FACBC IN PROSTATE CANCER: A SYSTEMATIC REVIEW AND META-ANALYSIS
The authors are thankful to the Reviewers for their comments. As requires, some updated information about the comparison of 68Ga-PSMA and 18F-FACBC has been added. Moreover, the English has been revised and other improvements have been given.
Please below you can find the point-by-point response (in red) to the queries.
REVIEWER 2
18F-FACBC is an FDA approved radiopharmaceutical for prostate cancer imaging. It is anticipated to be more prevalent in our clinical practice due to its potential to patients with biochemical recurrence following treatment. One of the concerns about the application of 18F-FACBC is its moderate specificity/performance at low PSA cases. Extensive studies in association with other feasible options are needed to validate this point. This manuscript is a systematic review and a meta-analysis to investigate the sensitivity and specificity of 18F-FACBC at the patient-based and regional-based levels. In my review, the provided list of publications on this is informative while the main questions about applications of 18F-FACBC cannot be answered due to lack of enough data to be analyzed.
Q2. My major concern about this paper is similarly of this study and its outcomes with the following papers:
1- Ren, Jingyun, et al. "The value of anti-1-amino-3-18F-fluorocyclobutane-1-carboxylic acid PET/CT in the diagnosis of recurrent prostate carcinoma: a meta-analysis." Acta radiologica 57.4 (2016): 487-493.
2- Yu, Chung Yao, et al. "Comparative performance of PET tracers in biochemical recurrence of prostate cancer: a critical analysis of literature." American journal of nuclear medicine and molecular imaging 4.6 (2014): 580.
None of the above paper has been cited in this manuscript and the “Discussion” section is only a list of highlighted information.
R2. Information provided by the suggested above mentioned manuscripts has been added, in order to reinforce the literature evidences. Please see the discussion paragraph that has been significantly improved.
Reviewer 3 Report
The manuscript entitled “18F-FACBC in Prostate Cancer: A Systematic review and meta-analysis” describes the review and meta-analysis to analyze the utility of 18F-FACBC in detection of primary/recurrent Prostate cancer and to investigate its diagnostic performance in positron emission tomography/computed-tomography (PET/CT). The analysis of the data and presentation is appropriate and the review will be useful to the readers. The review is acceptable after some minor modifications.
Author Response
CANCERS-559403
Tile manuscript: 18F-FACBC IN PROSTATE CANCER: A SYSTEMATIC REVIEW AND META-ANALYSIS
The authors are thankful to the Reviewers for their comments. As requires, some updated information about the comparison of 68Ga-PSMA and 18F-FACBC has been added. Moreover, the English has been revised and other improvements have been given.
Please below you can find the point-by-point response (in red) to the queries.
REVIEWER 3
The manuscript entitled “18F-FACBC in Prostate Cancer: A Systematic review and meta-analysis” describes the review and meta-analysis to analyze the utility of 18F-FACBC in detection of primary/recurrent Prostate cancer and to investigate its diagnostic performance in positron emission tomography/computed-tomography (PET/CT). The analysis of the data and presentation is appropriate and the review will be useful to the readers.
Q3. The review is acceptable after some minor English modifications.
R3. Some English revisions have been made.
Reviewer 4 Report
This is the first analysis for its kind. The number of the studies included are low with very high heterogeneity in several cases when pooled, and an updated analysis would be expected in the future as more FACBC scans will be done. Radiation oncology is currently the main source for ordering the FACBC scans. As time goes by, the updated review would have a block of patients from radio-therapy, which is lacking in this analysis.
The issues with this manuscript are,
Far more patients had PSMA-11 scans than FACBC scans. There are several recent published systemic reviews and/or meta-analyses on the use of PSMA-11, which were not discussed in current manuscript. Even though FACBC is the focus, a brief summary of the PSMA-11 analyses need to be included as least in the Discussion part of the manuscript. Similarly, many Choline PET scans were performed for prostate cancer before the rise of PSMA radio-tracers. There are published systemic and meta-analyses, even the comparison between the two, or three-way (choline, FACBC and PSMA-11) analysis. A brief review of the outcome fo those analyses in the Discussion is needed as well. The relationship between PET/CT vs PET/MRI vs mpMRI is complicated. There are not enough studies to compare two-ways or three-ways for any conclusions to be made. Combined use of these modalities will be another complication, which is rare currently. Schuster, Goodman, et al., are the developers of FACBC. Perhaps, their studies shall be excluded?!Author Response
CANCERS-559403
Tile manuscript: 18F-FACBC IN PROSTATE CANCER: A SYSTEMATIC REVIEW AND META-ANALYSIS
The authors are thankful to the Reviewers for their comments. As requires, some updated information about the comparison of 68Ga-PSMA and 18F-FACBC has been added. Moreover, the English has been revised and other improvements have been given.
Please below you can find the point-by-point response (in red) to the queries.
REVIEWIER 4
This is the first analysis for its kind. The number of the studies included are low with very high heterogeneity in several cases when pooled, and an updated analysis would be expected in the future as more FACBC scans will be done. Radiation oncology is currently the main source for ordering the FACBC scans. As time goes by, the updated review would have a block of patients from radio-therapy, which is lacking in this analysis.
The issues with this manuscript are, far more patients had PSMA-11 scans than FACBC scans. There are several recent published systemic reviews and/or meta-analyses on the use of PSMA-11, which were not discussed in current manuscript.
Q4. Even though FACBC is the focus, a brief summary of the PSMA-11 analyses need to be included as least in the Discussion part of the manuscript.
R4. A new paragraph about the comparison between FACBC-PET and PSMA-PET has been added as a sub-part of the discussion paragraph.
Q5. Similarly, many Choline PET scans were performed for prostate cancer before the rise of PSMA radio-tracers. There are published systemic and meta-analyses, even the comparison between the two, or three-way (choline, FACBC and PSMA-11) analysis. A brief review of the outcome for those analyses in the Discussion is needed as well.
R5. The discussion paragraph has been improved in the content, by mentioning data about the comparison between Choline and FACBC, as required.
Q6. The relationship between PET/CT vs PET/MRI vs mpMRI is complicated. There are not enough studies to compare two-ways or three-ways for any conclusions to be made. Combined use of these modalities will be another complication, which is rare currently.
R6. We agree with the comments of the Reviewer, as already stated in the discussion. In the vision of a complete analysis, we have added the following sentence: “However, the interpretation of these sophisticated imaging required a great experience and a significant learning curve”.
Q7. Schuster, Goodman, et al., are the developers of FACBC. Perhaps, their studies shall be excluded?!
R7. We do not agree with the exclusion of the studies by Schuster and Goodman, being their included in the systematic literature search.
Round 2
Reviewer 2 Report
In this revision, two new paragraphs has been added to the discussion section to address my concern about the previously published review articles about the same topic.
First, in the introduction section, this reviews should be mentioned. Then it should be followed with a sentence explaining the necessity of a new review.
Also, some parts of the manuscript is difficult to follow such as page 12:
"However, in the recent years, also PSMA-PET has rapidly been introduced in the flow-chart of patients with PCa, particularly in patients with a very early recurrence of disease (for a PSA levels < 1 ng/mL)."
The aforementioned lines need proofreading and a relevant citation. Also, the region-based outcomes of the previous reviews should be compared with your new data.
Author Response
In this revision, two new paragraphs has been added to the discussion section to address my concern about the previously published review articles about the same topic.
Q1. First, in the introduction section, this reviews should be mentioned. Then it should be followed with a sentence explaining the necessity of a new review.
R1. The following sentences have been added in the introduction paragraph: “Until now, few pooled data have been published about the role of 18F-FACBC PET/CT in patients with PCa. Ren et al () collected data from 6 studies, published between 2011 and 2014 and including 251 patients that concluded for a good sensitivity of 18F-FACBC PET/CT for the detection of PCa recurrence. In 2015, Yu et al () published a critical analysis of the available tracers for PET/CT in PCa, collecting data for 18F-FACBC from 5 studies (n=84 subjects), showing a limited detection rate of this imaging technique for the recurrence of post-prostatectomy PCa (detection rate=40%). However, in May 2016, 18F-FACBC PET/CT received the approval by the Food and Drug administration for use in patients with suspected recurrent PCa (No authors. FDA Approves 18F-Fluciclovine and 68Ga-DOTATATE Products. J Nucl Med. 2016;57:9N). In the last years, many retrospective and prospective experiences have been performed, and therefore an new update of the recent findings seems necessary, not only in the restaging but also in the initial staging of disease. Therefore….”
Q2. Also, some parts of the manuscript is difficult to follow such as page 12: "However, in the recent years, also PSMA-PET has rapidly been introduced in the flow-chart of patients with PCa, particularly in patients with a very early recurrence of disease (for a PSA levels < 1 ng/mL)." The aforementioned lines need proofreading and a relevant citation.
R2. The sentence has been changed as follow and a new reference has been added: “However, in the last years, PSMA-PET has rapidly been introduced in clinical practice, for the management of patients with recurrent PCa, particularly in case of low PSA levels (Mottet N, Bellmunt J, Bolla M, Briers E, Cumberbatch MG, De Santis M, Fossati N, et al. EAU-ESTRO-SIOG Guidelines on Prostate Cancer. Part 1: Screening, Diagnosis, and Local Treatment with Curative Intent. Eur Urol. 2017; 71:618-629.)”
Q3. Also, the region-based outcomes of the previous reviews should be compared with your new data.
R3. Data about the comparison region-based between the previous data and the current one has been added, only for the study by Yu et al (not being available for the study by Ren et al): “In the analysis by Yu et al (ref), FACBC showed a detection rate ranged between 22% and 61% for prostatic disease and between 19% and 33% for extra-prostatic disease, in accordance with the primary treatments (radical prostatectomy or radiotherapy). In our meta-analysis, we do not evaluated the pooled detection rate, but we calculated the pooled sensitivity and specificity. As illustrated in Table 4, the sensitivity of 18F-FACBC was equal to 90% for the identification of disease in the prostatic bed and 77% for extra-prostatic organs”.
Reviewer 4 Report
The authors have included the reports on PSMA-11 PET imaging in discussion. Readers will make their judgement.
However, it might be better to point out that the early publications are from the developers of FACBC.
Author Response
The authors have included the reports on PSMA-11 PET imaging in discussion. Readers will make their judgement.
Q1. However, it might be better to point out that the early publications are from the developers of FACBC.
R1. The following sentence has been added: “Also the papers by the developers of 18F-FACBC were considered (ref).”.
Round 3
Reviewer 2 Report
Regarding the positive feedback from other reviewers and the efforts of the authors in the improvement of this submission during revisions, I believe the paper could be considered for publication after following minor modifications:
1- Authors should use uppercase fonts for the atomic numbers of radioisotopes. For example:18F
2- Authorship should be according to MDPI guidelines.
3- I cannot see the name of the corresponding author of this manuscript on the first page.
4- “on the behalf of Young AIMN Working Group” is now listed in authors list which it is not clear if it refers to all or only the last author. In both cases, it should be as a footnote unless the editor makes an exception.
5- Abstract: Line # 5 PCa.A -> PCa. A
6- Table S1: the smilies should be replaced with a text, or you should add a footnote to define each colour-coded icon.
Author Response
Q1. Regarding the positive feedback from other reviewers and the efforts of the authors in the improvement of this submission during revisions.
R1. Many thanks for the comments.
I believe the paper could be considered for publication after following minor modifications:
Q2. Authors should use uppercase fonts for the atomic numbers of radioisotopes. For example:18F
R2. OK, the corrections have been made, in each part of the manuscript.
Q3. Authorship should be according to MDPI guidelines.
R3. The authorship has been modified, as requested.
Q4. I cannot see the name of the corresponding author of this manuscript on the first page.
R4. The name of the corresponding author has been added in the first page.
Q5. “on the behalf of Young AIMN Working Group” is now listed in authors list which it is not clear if it refers to all or only the last author. In both cases, it should be as a footnote unless the editor makes an exception.
R5. We have included a sentence, in order to avoid any mistake for the authorship.
Q6. Abstract: Line # 5 PCa.A -> PCa. A
R6. We have corrected it.
Q7. Table S1: the smilies should be replaced with a text, or you should add a footnote to define each colour-coded icon.
R7. A footnotes have been added.